# Current State of the Art and Prospects of T Cell-Redirecting Bispecific Antibodies in Multiple Myeloma

**DOI:** 10.3390/jcm10194593

**Published:** 2021-10-06

**Authors:** Mashhour Hosny, Christie P. M. Verkleij, Jort van der Schans, Kristine A. Frerichs, Tuna Mutis, Sonja Zweegman, Niels W. C. J. van de Donk

**Affiliations:** Cancer Center Amsterdam, Department of Hematology, Vrije Universiteit Amsterdam, Amsterdam UMC, 1081 HV Amsterdam, The Netherlands; m.hosny@amsterdamumc.nl (M.H.); c.verkleij@amsterdamumc.nl (C.P.M.V.); j.vanderschans@amsterdamumc.nl (J.v.d.S.); k.frerichs@amsterdamumc.nl (K.A.F.); t.mutis@amsterdamumc.nl (T.M.); s.zweegman@amsterdamumc.nl (S.Z.)

**Keywords:** bispecific antibody, multiple myeloma, BCMA, GPRC5D, CD38, FcRH5

## Abstract

Multiple myeloma (MM) patients eventually develop multi-drug-resistant disease with poor survival. Hence, the development of novel treatment strategies is of great importance. Recently, different classes of immunotherapeutic agents have shown great promise in heavily pre-treated MM, including T cell-redirecting bispecific antibodies (BsAbs). These BsAbs simultaneously interact with CD3 on effector T cells and a tumor-associated antigen on MM cells, resulting in redirection of T cells to MM cells. This leads to the formation of an immunologic synapse, the release of granzymes/perforins, and subsequent tumor cell lysis. Several ongoing phase 1 studies show substantial activity and a favorable toxicity profile with BCMA-, GPRC5D-, or FcRH5-targeting BsAbs in heavily pre-treated MM patients. Resistance mechanisms against BsAbs include tumor-related features, T cell characteristics, and impact of components of the immunosuppressive tumor microenvironment. Various clinical trials are currently evaluating combination therapy with a BsAb and another agent, such as a CD38-targeting antibody or an immunomodulatory drug (e.g., pomalidomide), to further improve response depth and duration. Additionally, the combination of two BsAbs, simultaneously targeting two different antigens to prevent antigen escape, is being explored in clinical studies. The evaluation of BsAbs in earlier lines of therapy, including newly diagnosed MM, is warranted, based on the efficacy of BsAbs in advanced MM.

## 1. Introduction

Multiple myeloma (MM) is the second most common hematological malignancy and is responsible for 2.1% of all cancer deaths in the U.S., as of 2020 [1]. MM is characterized by the clonal expansion of malignant plasma cells in the bone marrow, or less frequently in extramedullary sites [1,2]. Patients with MM suffer from end-organ damage, such as hypercalcemia, renal insufficiency, anemia, and/or bone disease with lytic lesions, which are known as CRAB features [2,3].

For many years, only classic chemotherapeutic agents (e.g., melphalan, cyclophosphamide, and anthracyclines) and glucocorticosteroids (dexamethasone and prednisone) were available for the treatment of MM [4]. In the last two decades, several novel drugs were introduced, such as immunomodulatory drugs (IMiDs; thalidomide, lenalidomide, and pomalidomide), histone deacetylase inhibitors, proteasome inhibitors (PIs; bortezomib, ixazomib, and carfilzomib), and naked CD38- or SLAMF7-targeting monoclonal antibodies (mAbs; daratumumab, isatuximab, and elotuzumab) [3,5,6]. Most recently, the incorporation of CD38-targeting antibodies into both first-line and relapse regimens has substantially improved the progression-free survival (PFS) and overall survival (OS) of both newly diagnosed and relapsed/refractory (R/R) MM patients [7,8,9,10,11]. Although these novel drugs have significantly improved the outcome of MM, the majority of patients will eventually develop multi-drug-resistant disease, which is associated with very poor survival [12,13]. Managing late-stage R/R MM still represents a significant challenge in clinical practice [14]. This underscores the urgency to identify novel treatment strategies, which can effectively target multi-drug-resistant MM clones.

In the last few years, novel immunotherapeutic formats were developed and evaluated in heavily pre-treated patients. This has recently led to new approvals for this subset of patients, including the BCMA-targeting chimeric antigen receptor (CAR) T cell product ide-cel (Abecma) and the antibody-drug conjugate belantamab mafodotin (Blenrep), a BCMA-targeting mAb conjugated to the cytotoxic agent monomethyl auristatin-F [4,15,16,17]. In addition, several new antibody formats were developed, including T cell-redirecting bispecific antibodies (BsAbs). These BsAbs have two binding domains enabling simultaneous interaction with CD3 on effector T cells and with a tumor-associated antigen (TAA), resulting in the redirection of T cells to the tumor cells and subsequent formation of an immunologic synapse (Figure 1). This is followed by T cell activation and degranulation, with the release of granzymes and perforins, and eventually tumor cell lysis [18,19,20]. Importantly, BsAbs induce T cell activation independent of antigen presentation on the major-histocompatibility complex (MHC) class 1 [21,22]. Additionally, BsAbs are capable of initiating T cell activation without the need for co-stimulation, and are therefore independent of antigen-presenting cells or cytokines [23,24,25,26]. Clinical studies with various BsAbs have recently demonstrated promising activity with a favorable toxicity profile in heavily pre-treated MM patients (Table 1) [14,26,27].

In this review, we provide an overview of the different BsAb formats and the various BsAbs, targeting different MM-associated antigens, which are currently evaluated in clinical studies. We will also discuss the challenges of BsAbs, for instance, emerging resistance mechanisms. Finally, the future direction of BsAbs as an MM treatment will be addressed.

## 2. Bispecific Antibodies: Overview and Characteristics of Different BsAbs Formats

The graft-versus-myeloma effect mediated by donor T cells following allogeneic stem cell transplantation (allo-SCT) or donor lymphocyte infusions (DLIs) has demonstrated the potency of T cells to eliminate MM cells [14,28]. Unfortunately, allo-SCT and DLIs are also associated with considerable toxicities, such as graft-versus-host disease and infections, which may result in treatment-related mortality in a substantial number of patients [14,28]. Therefore, novel therapeutic strategies have been developed that use the killing capacity of T cells in a more tumor-specific way to improve the toxicity profile. First, T cells can be reprogrammed to target MM cells by introducing genes encoding CARs [29]. Second, T cells can be redirected to tumor cells by bispecific T cell engagers (BiTEs) and T cell redirecting BsAbs (Figure 1).

Although there are different BsAb formats, at this moment, two classes of BsAb are predominantly evaluated in clinical trials. This includes IgG-like BsAbs consisting of fragment antigen-binding (Fab) domains connected to a fragment-crystallizable (Fc) region, and BiTEs consisting of two scFv units fused with a short flexible linker (Figure 1) [14,18,19,26,30]. The presence of an Fc domain in IgG-like immunoglobulin increases the size and stability of BsAbs, which leads to an extended half-life [14,18,19,26,30]. The Fc domain can also be used to initiate Fc-mediated effector functions, such as antibody-dependent cell-mediated cytotoxicity (ADCC) and complement fixation [14,18,19,26,30]. However, T cell-redirecting BsAbs are engineered to have an effector-silenced Fc region, thereby preventing excessive cytokine secretion [31]. One of the advantages of BiTEs includes the relative ease of changing or adding another scFv to create a trivalent molecule, but they also have drawbacks compared to intact BsAbs, such as the short half-life due to their small sizes and therefore the need for continuous infusion [18,26,27,30,32,33,34].

Trispecifc antibodies (TsAbs) are capable of engaging three different antigens (Figure 1). These molecules can target two antigens on the tumor cell, resulting in more efficient T cell redirection to tumor cells, and possibly the prevention of development of antigen escape variants [35]. Alternatively, trispecific antibodies can simultaneously engage one tumor antigen and two antigens on T cells, leading to improved T cell signal transduction, while targeting the T cells specifically to tumor cells [35].

## 3. BsAb MM Targets and Their Potential as a Therapeutic Agent

Because of the high potency of BsAbs, the target antigen should fulfil various characteristics. To prevent toxicity and antigen escape, the tumor-associated antigen should be widely and homogeneously expressed on malignant cells and not on normal tissues [26]. In addition, the target antigen should be critical for the tumor cell so that down-regulation or complete loss of target expression will be disadvantageous for the tumor. [46]. Noteworthy, antigen distribution and expression may vary within the tumor and between patients. Target internalization may also have a negative impact on the activity of the BsAb, as this will limit immune synapse formation. In addition, the activity of BsAbs is also affected by their molecular format, including the binding affinity of each arm. Furthermore, for antigens with large extracellular domains, the epitope targeted by the antibody may influence the efficiency of immune synapse formation, with BsAbs binding to the membrane proximal region having more efficient immune synapse formation, compared to BsAbs binding the central or distal domains [47]. Here, we provide an overview of the BsAbs under clinical evaluation in MM, targeting a tumor-associated antigen and the CD3-subunit of the T cell receptor.

### 3.1. B Cell Maturation Antigen (BCMA)

BCMA, also known as CD269 or TNFRSF17, is exclusively expressed in a subset of mature B cells and normal and malignant plasma cells (PCs). This antigen is currently the most frequently used target for T cell redirection therapy in MM [48]. BCMA promotes plasma cell survival and thereby supports long-term humoral immunity, but it is not involved in early B cell development and overall B cell homeostasis [26,49,50,51]. BCMA is a type III transmembrane glycoprotein and a member of the tumor necrosis family receptor (TNFR) superfamily. It interacts with its ligand B cell activating factor (BAFF) and a proliferation-inducing ligand (APRIL) [51]. This interaction results in downstream signals to TNF-R-associated factor 1 (TRAF-1), TRAF-2, and TRAF-3, inducing activation of NF-κB, ETS-like transcription factor 1 (Elk-1), c-jun N-terminal kinase (JNK), and p38 [52]. This leads to the enhancement of B cell survival and proliferation through upregulation of anti-apoptotic proteins, such as Bcl-2 and Bcl-XL, and plasma cell differentiation [53,54]. The enzyme complex γ-secretase cleaves BCMA off the cell membrane, releasing soluble BCMA (sBCMA), which can subsequently neutralize APRIL [55]. Inhibition of γ-secretase enhanced BCMA surface expression on plasma cells and promoted clonal expansion in vivo [55]. Studies found elevated sBCMA in the serum of MM patients correlating with a more advanced state of the disease and shorter overall survival [56,57,58]. sBCMA can also serve as a novel biomarker for response in MM patients [57].

#### 3.1.1. BsAbs Targeting BCMA Evaluated in Patients

##### Pacanalotamab (AMG 420)

AMG420 is the first-in-class BCMA-specific BiTE antibody that is comprised of two scFvs linked by a flexible linker targeting both BCMA and CD3 [34]. In preclinical studies, it was found that AMG420 was able to induce T cell-mediated lysis of BCMA^+^ MM cells, without effects on BCMA^-^ cells [26,59]. AMG420 also induced lysis of primary MM cells in samples from newly diagnosed and R/R MM patients. It also showed dose-dependent anti-tumoral activity in MM mouse models [59]. In addition, dose-dependent depletion of malignant BCMA^+^ PCs was observed in the bone marrow of cynomolgus monkeys after administration of AMG420. Importantly, clinical signs of toxicity remained limited and cytokine levels reached normal values at 24 h post either intravenous (IV) or subcutaneous (SC) administration at dose levels ≥15 µg/kg/day [59].

Altogether, based on these preclinical findings, a phase 1 first-in-human dose-escalation and expansion study (NCT02514239) was initiated with R/R MM patients. In this study, AMG420 was administered as a continuous intravenous infusion during 4 weeks of each 6-week cycle, because of its short half-life [34]. This trial enrolled 42 patients, who had disease progression after ≥2 lines of prior therapy (median of 3.5 prior lines of therapy). Patients with extramedullary disease were excluded. AMG420 induced an overall response rate (ORR) of 31% (13 of 42 patients), which was superior (70%; 7 of 10 patients) in the patients treated at the maximum tolerated dose (MTD) of 400 µg/day [34]. Of the seven responding patients treated at the MTD, five achieved a minimal residual disease (MRD)-negative complete remission, one had a partial response (PR), and one a very good partial response (VGPR). During the trial, 38% of the patients experienced cytokine release syndrome (CRS; 16 of 42 patients; grade 1 28%, grade 2 5%, grade 3 2%). Infections were observed in 14 of 42 patients (33%) [34]. Other treatment-emergent adverse events (TEAEs) included one patient with grade 3 edema and 2 patients with grade 3 polyneuropathy [34]. Despite promising efficacy, clinical development of AMG420 was terminated because of practical problems associated with the continuous intravenous infusion [27].

##### Pavurutamab (AMG 701)

Because of the practical issues in terms of the continuous intravenous infusion of AMG420, a half-life extended (HLE) BiTE antibody (AMG701) was developed. Like AMG420, this antibody is also comprised of anti-BCMA and anti-CD3 scFvs, but with the addition of an HLE Fc domain [19,26,60]. AMG701 has a half-life of approximately five days in non-human primates, enabling once-weekly administration [60]. In preclinical studies with cell lines and patients’ samples, AMG701 was able to induce T cell-dependent cellular cytotoxicity and dose-dependent T cell activation, without effects on BCMA^-^ MM cells, indicating the specificity of this HLE BiTE molecule [26,61,62]. AMG701 also induced T cell proliferation and differentiation towards a central and effector memory phenotype [61,62]. In addition, in vivo experiments with NOD/SCID mice showed inhibition of tumor growth and extended survival with AMG701. Furthermore, AMG701 led to a depletion of bone marrow-localized PCs in cynomolgus monkeys.

Currently, AMG701 is being tested in a dose-escalation and expansion phase 1 trial (NCT03287908). Up till now, 85 patients with R/R MM were enrolled [36]. These patients had disease progression after ≥3 lines of prior therapy (median of 6 prior lines of therapy, 93% triple-exposed, 62% triple-refractory), and patients treated with anti-BCMA agents and stem cell transplant within 3–6 months were excluded [36]. Patients were treated with weekly IV AMG701 infusions until disease progression (PD). From the most recent evaluable cohort, AMG701 treatment resulted in an ORR of 83% (5 of 6 patients). Of the seven patients that were tested for MRD, six were MRD negative (at the level of 10^−5^) and all six patients had ongoing responses, up to 22 months in one patient [36]. CRS was common and mostly grade 1 (27%; *n* = 23) and grade 2 (28%; *n* = 24). Eight patients (9%) experienced grade 3 CRS, which improved with corticosteroids and tocilizumab (median CRS duration: 2 days). PK assessments showed that AMG701 exposures increased in a dose-related manner [36].

##### Elranatamab (PF-06863135)

PF-3135 is a humanized IgG-like bispecific monoclonal antibody (IgG2a) that utilizes two arms to target BCMA and CD3. In preclinical studies, elranatamab was able to kill BCMA^+^ MM tumor cells in vitro and in a MM mouse model [63]. Elranatamab was also capable of completely depleting BCMA^+^ PCs in cynomolgus monkeys with a half-life of 4–6 days [26,63]. Based on these preclinical data, elranatamab is currently under clinical investigation in a phase 1 dose-escalation study (NCT03269136). In this study, weekly IV administration of PF-3135 (0.1–50 µg/kg) was effective in R/R MM patients who had received ≥3 lines of prior therapy^51^. Although IV administration of elranatamab showed an acceptable tolerability profile, weekly SC administration was subsequently assessed [37,64]. Until now, no dose-limiting toxicities (DLTs) have been observed across all evaluated dose levels (80–1000 μg/kg weekly) in the 30 enrolled R/R MM patients (median of 8 prior lines of therapy, 87% triple-class refractory) of which seven patients had received prior BCMA-targeted therapy [37,65]. The most common grade ≥ 1 TEAEs included CRS (73%), anemia (60%), injection site reaction (50%), thrombocytopenia (53%), and neutropenia (53%). There were no cases of grade 3 or higher CRS (grade 1: 57%; grade 2: 17%) [37]. Subcutaneous administration of elranatamab resulted in clinical responses beginning at 215 µg/kg with an ORR of 70% (14 of 20 patients), including very good partial response (VGPR) in 35%, and complete response (CR)/stringent complete response (sCR) in 30% [37]. At the RP2D of 1000 µg/kg, the ORR was 83% [37]. Interestingly, activity was also observed in three of four patients with prior BCMA-directed therapy (2 VGPR, 1 sCR). All three MRD-evaluable patients were MRD negative (at the level of 10^−6^) [37]. For the 14 responding patients, the median duration of response has not yet been reached (92.3% of patients were free of progression at 6 months). Importantly, SC dosing of elranatamab resulted in a prolonged absorption phase without observing an increased severity of CRS compared to IV dosing.

##### Teclistamab (JNJ-64007957)

Teclistamab is a humanized BCMA x CD3 full-size bispecific antibody with an IgG4 Fc region [38]. Assessment of preclinical activity of teclistamab showed potent antitumor activity against various MM cell lines and elimination of malignant PCs from 55 patient-derived BM samples in a dose-dependent manner [66,67]. MM cell elimination was accompanied by activation and degranulation of CD4^+^ and CD8^+^ T cells, as well as their production of proinflammatory cytokines [66,67]. Teclistamab was also investigated in MM mouse models and showed significant activity. Furthermore, teclistamab was well-tolerated by cynomolgus monkeys when treated with doses up to 10 mg/kg/week for 5 weeks and exposure was dose proportional [68]. Hence, this BsAb was further investigated in a phase 1 dose-escalation study (NCT03145181) where teclistamab was given intravenously or subcutaneously and the RP2D has now been identified as a weekly subcutaneous administration of 1500 µg/kg [38,69]. In the most recent update of the results, 40 patients were treated at the RP2D (median of 5 prior lines of therapy, 83% triple-class refractory, 38% penta-drug refractory). These patients were treated with step-up doses of 60 and 300 µg/kg to mitigate CRS. Teclistamab was well tolerated and no DLTs were observed in RP2D-dosed patients [38]. CRS events (grade 1 = 45%; grade 2 = 25%; no grade ≥ 3 events) occurred in 70% of the RP2D-dosed patients and resolved in all patients. In this subset of 40 patients, the ORR was 65% (≥VGPR: 58%; ≥CR: 40%) with a median time to first confirmed response of 1.0 month. The ORR in 33 triple-class refractory patients was 61% [38]. Moreover, all six MRD-evaluable patients in the RP2D cohort were MRD negative (at the level of 10^−6^ in 5, and 10^−6^ in 1). Across IV and SC cohorts, 18/26 (69%) evaluable patients who achieved a CR were MRD negative (at the level of 10^−6^ in 16, or 10^−5^ in 2) [38]. After a median follow-up of 7.1 months, 22/26 responders (85%) were alive and continuing treatment. The 6-month PFS rate in patients treated at the RP2D was 67% [69]. Based on the current data, an expansion study of teclistamab at the RP2D in patients with R/R MM is ongoing (NCT04557098).

##### REGN5458

REGN5458 is a fully human BsAb with two binding sites targeting BCMA and CD3, and an Fc domain without effector function [70]. In vitro studies demonstrated that REGN5458 induced efficient lysis of various MM cell lines with a range of different BCMA expression levels, and primary tumor cells [70]. In immunodeficient NSG mice, REGN5458 led to a dose-dependent inhibition of tumor growth [70]. Depletion of BCMA^+^ PCs was also observed in cynomolgus monkeys treated with this BsAb. Moreover, REGN5458 was compared to an anti-BCMA CAR T cell with an scFv-based CAR construct reformatted from the anti-BCMA arm of REGN5458, including CD8 hinge/transmembrane, 4-1BB co-stimulatory, and CD3ζ signaling domains. Similar antitumor activity was observed against high- and low-BCMA-expressing cells and MM patient-derived cells in vitro [70]. Comparable antitumor activity was also observed in an MM mouse model, but the CAR T cells with 4-1BB/CD3ζ signaling domains exhibited slower anti-tumor kinetics [70]. CAR T cells with a CD28 costimulatory domain exhibited more rapid anti-tumor kinetics [70].

Currently, REGN5458 is being investigated in a phase 1 dose-escalation study (NCT03761108) with R/R MM patients (*n* = 49) that received ≥3 prior lines of therapy (median of 5 prior lines of therapy, 100% triple-class refractory, 57% penta-drug refractory) [40]. REGN5458 was administered once weekly as IV infusion, followed after 16 weeks by a once every 2 weeks dosing. In this study, two patients experienced a DLT: one patient with acute kidney injury (grade 4, dose level 4 = 24 mg), and one patient with grade 3 elevated aspartate transaminase (AST)/alanine aminotransferase (ALT) (dose level 6 = 96 mg). Across all dose levels, the most common grade ≥ 3 AEs were anemia (22%), thrombocytopenia (6%), and neutropenia (14%). CRS was observed in 39% of patients. Importantly, no grade ≥ 3 events were reported (grade 1 CRS in 33% and grade 2 CRS in 6%). REGN5458 treatment induced an ORR of 39% (19 of 49 patients); 95% (18 of 19 patients) of the responders achieved VGPR or better, including 42% (8 of 19 patients) with CR or sCR. Four out of seven evaluable patients (57%) achieved MRD negativity (at the level of 10^−5^) [40]. The median duration of response was 6 months [40]. Importantly, REGN5458 treatment resulted in a significant improvement in quality of life and pain symptoms [71].

In addition, REGN5459, another BCMA-targeting BsAb with different binding characteristics compared to REGN5458, is now being evaluated in a phase 1 dose-escalation study (NCT04083534). There are currently no clinical data available.

##### TNB-383B

TNB-383B is a fully human trivalent BsAb that consists of two anti-BCMA domains and one anti-CD3 moiety in a silenced IgG4 backbone [72]. After assessing 12 different anti-CD3 antibodies, one unique CD3-targeting arm was chosen based on effective MM cell lysis accompanied by low levels of cytokine secretion [72]. In addition, anti-tumor activity was observed in MM mouse models [72,73]. Based on these findings, TNB-383B is currently being investigated in a phase 1 dose-escalation and expansion trial (NCT03933735), which up till now enrolled 58 patients with ≥3 prior lines of therapy (median of 6 prior lines of therapy, 64% triple-class refractory, 34% penta-drug refractory). TNB-383B was intravenously administered once every three weeks (Q3W). TNB-383B was well tolerated, with 45% (26 of 58 the patients) of the patients experiencing CRS [41]. At the highest dose levels evaluated up till now (40–60 mg), 80% of the patients experienced CRS (grade 1 in 47%, grade 2 in 33%, no grade ≥ 3 events). The most common grade ≥ 3 TEAEs were anemia (17%), neutropenia (16%), thrombocytopenia (14%), and infections (14%). Across all doses, TNB-383B induced an ORR of 46.5%, which was 80% at doses ≥40 mg/dose, including ≥VGPR in 73% [41]. Four subjects were MRD-evaluable, of which three became MRD negative at the level of 10^−6^ (*n* = 2) or 10^−5^ (*n* = 1) [41]. At the last data cut-off, 22/27 (81%) of the responders showed an ongoing response (up to 66 weeks) [41].

##### Alnuctamab (CC-93269 or EM901)

CC-93269 (EM901) is another humanized trivalent BsAb that is under clinical investigation in R/R MM. To increase the avidity, this BsAb has a bivalent anti-BCMA arm and a monovalent anti-CD3 domain, both connected to a silenced IgG1-based Fc region [74]. It is currently being investigated in a phase 1 dose-escalation and expansion study (NCT03486067) with patients who had previously received ≥3 prior regimens (median of 5 prior lines of therapy, 67% triple-class refractory), with prior BCMA-directed therapy not allowed. In the most recent update from the phase 1 study, 30 patients were treated with IV CC-93269, administered weekly during the first 3 28-day cycles, and then biweekly during cycles 4–6, which was followed by once every 4 weeks for up to 2 years [42]. Similar to other BsAbs, response to CC-93269 is dose dependent. Overall, 13 patients (43%) achieved ≥PR, which included CR/sCR in 5 (17%) patients [42]. Within the cohort of nine patients treated with the highest dose (10 mg) of CC-93269, the ORR was 89% with ≥CR in 44% of these patients [42]. The median time to response was 4.1 weeks [42]. Moreover, among the 13 responders, 92% achieved MRD negativity (at the level of 10^−5^). Safety assessment of CC-93269 showed that 29 of the 30 patients experienced at least one grade ≥ 1 TEAE, and one or more grade ≥ 3 TEAEs occurred in 22 patients (73%) [26,42]. The most common grade ≥ 1 TEAEs included neutropenia (47%), anemia (43%), infections (57%), and thrombocytopenia (30%) [26,42]. Twenty-three patients (77%) developed any grade CRS. The majority experienced grade 1 (50%) or grade 2 (23%) CRS, but one patient suffered from a grade 5 CRS after receiving 6 mg of CC-923269 as the first dose and 10 mg on day eight of the first cycle. The patient died during the study with contributing factors of a high tumor burden with extensive extramedullary disease and concomitant infections [42].

### 3.2. CD38

CD38 is a 45-KD, type II transmembrane glycoprotein of the ADP-ribosyl cyclase family [75,76]. CD38 has multiple functions, such as regulation of calcium homeostasis [11,26,75,76]. CD38 also regulates different signaling pathways that are involved in proliferation, growth, activation, and survival of lymphoid and myeloid cells [76]. As a receptor, CD38 interacts with the cell surface ligand CD31 that is expressed by endothelial cells to initiate the migration of leukocytes [77]. This interaction may also contribute to MM cell survival due to the adhesion of MM cells to BM endothelial and stromal cells [26]. CD38 is highly expressed on MM cells and normal PCs but is also expressed by a wide variety of other cell types, such as normal lymphoid cells, myeloid cells, and nonhematopoietic cells or tissue [26,75,76,77]. Based on the activity of naked CD38-targeting antibodies, such as daratumumab, BsAbs targeting CD38 are currently being investigated in clinical studies in R/R MM [7,9,11].

#### 3.2.1. BsAbs Targeting CD38 in the Clinic

##### AMG 424

AMG424 is a humanized CD38 x CD3 BsAb, consisting of a hetero-Fc domain devoid of Fcγ receptor and complement binding [24]. AMG424 was selected out of a panel of CD38 x CD3 BsAbs based on its ability to effectively eliminate cancer cells expressing high and low levels of CD38, but with attenuated cytokine release (because of relatively low CD3 binding affinity) [24]. However, in vitro AMG424 also triggered “off-tumor” cytotoxicity against CD38-expressing B, T, and NK cells. In addition, depletion of CD38-expressing cells and T cell activation was observed after intravenous administration of AMG424 in cynomolgus monkeys [24]. In MM mouse models, AMG424 inhibited tumor growth, which was accompanied by T cell activation [24]. Noteworthy, AMG424 administration also caused a decrease in CD4^+^ and CD8^+^ T cell numbers, which may be explained by in vivo T cell fratricide [24]. A phase 1 study (NCT03445663) with AMG424 was initiated to assess its safety and tolerability and to determine the MTD, but the study was terminated due to a sponsor business decision. Hence, no clinical data are available.

##### GBR 1342 (ISB 1342)

GBR 1342, or ISB 1342, is a novel CD38 x CD3 BsAb constructed using the BEAT^®^ platform [78]. BsAb has an scFv arm targeting CD3ε and a Fab arm, which specifically recognizes CD38 and does not compete with daratumumab [78]. The Fc tail was engineered to reduce CDC and ADCC effector functions. Preclinical studies showed potent killing of CD38^+^ cancer cell lines, whereby cell lysis was directly correlated with the level of target expression [78]. In in vitro assays with cell lines, GBR 1342 showed greater efficacy than daratumumab, and CD38^+^ T cell fratricide was not observed [78]. A phase 1 clinical trial is enrolling patients with prior exposure to PI, IMiD, and daratumumab (>6 months prior to enrollment) to determine the RP2D (NCT03309111) [43]. Clinical data on safety and efficacy are not yet available.

### 3.3. B Lymphocyte Antigen CD19 (CD19)

CD19 is a B lymphocyte-specific type I transmembrane glycoprotein that belongs to the immunoglobulin superfamily and is expressed on all B lineage cells [26,79,80,81]. CD19 plays a crucial role in modulating B cell receptor (BCR)-dependent and -independent signaling to establish the intrinsic B cell signaling threshold affecting cell survival [79,80,81]. In addition, a small population of CD19^+^ MM cells with a less differentiated phenotype and with disease-propagating properties can be identified with flow cytometry. Additionally, more sensitive techniques, such as dSTORM, have shown that CD19 is expressed in the majority of MM patients on a substantial fraction of tumor cells but at low or ultra-low levels [82]. The low levels of CD19 on the MM cell surface were able to trigger anti-CD19 CAR T cells to eliminate these tumor cells [82]. Moreover, several studies suggest that patients with R/R MM may benefit from anti-CD19 CAR T cell therapy, or from CAR T cells targeting both CD19 and BCMA [15]. Altogether, this suggests that CD19 is also an interesting potential target for BsAbs.

#### 3.3.1. BsAbs Targeting CD19 in the Clinic

##### Blinatumomab

Blinatumomab is approved by the FDA for the treatment of R/R B cell acute lymphoblastic leukemia (ALL) in adults and children [83]. It consists of two scFvs against both CD19 and CD3 in a BiTE format with a half-life of approximately 2 h in humans [84]. Due to its activity in R/R B-ALL, a phase I trial (NCT03173430) of blinatumomab after salvage autoSCT in patients with R/R MM was initiated [26,85]. However, this pilot study was terminated because of too slow accrual; hence, no clinical data are available [85]. However, there is a case study of a 70-year-old patient with R/R MM, as well as precursor B-ALL [33]. The patient achieved MRD-negative complete remission of the ALL and a VGPR of the MM after the first blinatumomab cycle. Importantly, MM cells derived from this patient were positive for CD19 by flow cytometry.

### 3.4. Fc Receptor-Homolog 5 (FcRH5)

FcRH5, also designated as CD307, is a membrane protein that is closely related to the family of receptors homologous to FcγRI [26,86,87,88]. FcRH5 is expressed on B cell lineage cells, and already detectable on pre-B cells. It is also highly expressed on plasma cells, at higher levels than on normal B cells [26,47]. Additionally, different B cell malignancies were found to express FcRH5 including MM cells with a near 100% prevalence of FcRH5 [47,89]. Reports show that FcRH5 can bind to intact and aggregated IgG [47,86,87]. However, its biological significance remains largely unclear. Interestingly, aggressive MM is associated with the amplification 1q21, which is also the location of the FcRH5 gene [90,91]. A significant association between elevated FcRH5 RNA expression and 1q21 gain was found after analyzing various primary MM samples [47]. Importantly, FcRH5 has a large extracellular region that may affect the efficiency of T cell synapse formation due to the increased distance between the epitope and the target cell membrane [92]. Therefore, BsAbs targeting a membrane-proximal epitope on FcRH5 may be more effective than antibodies that target more distal regions [47].

#### 3.4.1. BsAbs Targeting FcRH5 in the Clinic

##### Cevostamab (BFCR4350A)

Cevostamab is a humanized IgG1-based BsAb that targets a membrane-proximal extracellular domain of FcRH5 on MM cells as well as CD3 on T cells [47]. This facilitates more efficient synapse formation and improved killing activity of T cells against MM tumor cells [47]. In preclinical assays, cevostamab was able to mediate lysis of FcRH5-positive MM cells in a dose-dependent manner [47]. Cevostamab also induced proliferation of effector T cells in the presence of target cells [47]. Moreover, tumor regression was achieved with cevostamab in an MM mouse model [47]. In cynomolgus monkeys, cevostamab mediated potent killing of B cells and BM-localized PCs in a dose-dependent manner [47].

Currently, cevostamab is being investigated in a phase 1 dose-escalation and expansion study (NCT03275103). In the most recent update of the study, 53 patients (median of 6 prior lines of therapy, 72% triple-class refractory, 45% penta-drug refractory) were enrolled [44]. Prior treatment with CAR T cells, other BsAbs, and antibody-drug conjugates was allowed [44]. Cevostamab was administered as IV infusion every 3 weeks, up to a maximum of 17 cycles; the step-up dose ranged from 0.05 to 3.6 mg, and the target dose ranged from 0.15 to 160 mg. Responses were observed at doses ≥3.6/20 mg and within these cohorts, the ORR was 53% (18 of 34 patients; ≥VGPR: 32%). In the subset of penta-drug refractory patients, the ORR was 41% (7 of 17 patients), and in the subset of patients with prior anti-BCMA therapy, the ORR was 63% (5 of 8 patients) [44]. Across all cohorts, eight responding patients had a response duration of ≥6 months. MRD negativity (at the level of 10^−5^) was detected in six out of sevven evaluable patients, who had achieved at least a VGPR [44].

Safety assessment of cevostamab showed that CRS was the most common TEAE: 40/53 patients (76%) experienced any grade CRS, most common during the first cycle. Grade 1 CRS was observed in 34%, grade 2 CRS in 40%, and grade 3 CRS in 1/53 patients (2%) [44]. All CRS events resolved within two days. Other frequent ≥ grade 1 TEAEs were neutropenia (17%), thrombocytopenia (32%), and anemia (28%) [44]. One patient treated with 3.6/90mg cevostamab developed a DLT, characterized as a grade 3 pneumonia, but MTD was not reached [44].

### 3.5. G Protein-Coupled Receptor Family C Group 5 Member D (GPRC5D)

GPRC5D is a type-C 7-pass transmembrane orphan receptor protein, whereas its ligand and intracellular role are yet to be determined [93,94,95]. GPRC5D is highly expressed on MM cells, while GPRC5D expression was not observed on normal hematopoietic cells and BM progenitors, including hemopoietic stem cells [94,96]. In addition, GPRC5D is expressed in cells that produce hard keratin, such as cortical cells of the hair shaft and the keratogenous zone of the nail [97].

#### 3.5.1. BsAbs Targeting GPRC5D in the Clinic

##### Talquetamab (JNJ-64407564)

Talquetamab is a humanized IgG-like BsAb with an IgG4 Fc region that simultaneously targets GPRC5D^+^ and CD3^+^ T cells, resulting in the elimination of MM cells [93]. In vitro studies showed the ability of talquetamab to mediate potent killing of various MM cell lines with different levels of GPRC5D expression after co-incubation [20,93]. Talquetamab also induced, in a dose-dependent manner, T cell activation and degranulation, as well as increased cytokine secretion [20,93]. In addition, talquetamab mediated potent killing of MM cells derived from patients with newly diagnosed MM or R/R MM, which was accompanied by a significant increase in CD4^+^ and CD8^+^ T cell activation and degranulation [20]. Talquetamab also has significant antitumor activity in MM mouse models [93].

Based on promising preclinical data, a phase 1 dose-escalation and expansion study (NCT03399799) was initiated with talquetamab. As of 18 April 2021, a total of 184 patients had received talquetamab IV (*n* = 102) or SC (*n* = 82) [45]. In this study, the RP2D of talquetamab was identified as a weekly SC dose of 405 µg/kg, with 10.0 and 60.0 µg/kg step-up doses [93]. In the most recent update of the study, 30 patients were treated at the RP2D (median of 6 prior lines of therapy, which included prior anti-BCMA therapies in 8 patients (27%)). All patients were triple-class exposed, 77% were triple-class refractory, and 20% were penta-drug refractory [45]. Talquetamab was well tolerated and no DLTs were observed at the RP2D [45]. The most common TEAE at the RP2D was CRS (all grade CRS: 73%). CRS was limited to grade 1 (60%) and grade 2 (10%), with the exception of one patient (3%) who experienced grade 3 CRS [45]. CRS was generally confined to the step-up and first full dose. Other common TEAEs included neutropenia (all grade: 67%; grade 3/4: 60%), anemia (all grade: 57%; grade 3/4: 27%), and dysgeusia (all grade: 60%; grade 3/4: 0%) [45]. In addition, 32% of the patients treated at the RP2D developed infections, but only 3% were grade 3/4 infections. Skin-related AEs were observed in 77% of the patients, and nail disorders in 27% of the patients [45].

In the RP2D cohort, talquetamab resulted in an ORR of 70% (21 of 30 patients, ≥VGPR in 60%) [45]. Similar efficacy was seen in the subset of patients with triple-class refractory disease (ORR: 65.2%) and in patients with penta-drug refractory disease (ORR: 83.3%) [45]. Moreover, across both IV and SC cohorts, six patients were evaluable for MRD negativity of which four had an MRD-negative CR (at the level of 10^−6^), including one patient treated with talquetamab at the RP2D. Responses were durable and continued to deepen over time [45]. At the RP2D, the median duration of response was not reached, and 17/21 responders (81%) were continuing treatment after a median follow-up of 6.3 months. A phase 2 expansion study (NCT04634552) of talquetamab at the RP2D is ongoing.

## 4. Discussion

In this review, we show a favorable balance between the activity and tolerability of BsAbs in heavily pre-treated MM patients, often lacking alternative treatment options. It is expected that in the near future, several BsAbs will be approved for the treatment of patients with heavily pre-treated MM. Although BCMA is the most common target for BsAbs under clinical evaluation in MM, BsAbs targeting alternative antigens (e.g., FcRH5, and GPRC5D) are also promising.

Although responses are durable, patients will eventually relapse. A better understanding of the mechanisms of relapse is important as it may contribute to novel, more effective, combination strategies. Resistance is probably related to tumor-related features, T cell characteristics, and the immunosuppressive microenvironment (Figure 2). BCMA loss has recently been described as a cause of acquired resistance to a BCMA-specific BsAb [98]. Targeting more than one tumor-associated antigen may prevent the development of antigen escape variants. The combination of teclistamab and talquetamab is currently being evaluated in a clinical trial. Clinical studies are also evaluating the combination of BsAbs with IMiDs, which have T cell stimulatory effects and enhance the activity of BsAbs in preclinical studies [20,62,99]. Daratumumab is also an interesting combination partner for BsAbs, because it has the ability to eliminate Tregs and improves T cell numbers and activity. Indeed, we have shown that daratumumab enhanced the elimination of MM cells when it was combined with either teclistamab or talquetamab [20,66].

To what extent prior BCMA exposure impacts response to a BCMA-targeting BsAb is currently still an open question. However, in the Magnetismm-1 study, three of four elranatamab-treated patients with prior BCMA-directed therapy achieved a response (two VGPR and one sCR) [65]. Alternatively, patients who progress following BCMA-directed therapy can also be treated with a BsAb that targets another MM-associated antigen, such as GPRC5D [45] or FcHR5 [44]. Notably, response to cevostamab was not affected by prior exposure to a BCMA-targeting agent [100]. Importantly, because of the high activity and favorable toxicity profile of BsAbs in end-stage MM, several trials are now designed to evaluate the activity of BsAbs alone or in combination in patients with earlier stages of the disease, including early relapsed MM and newly diagnosed disease [27].

The BsAbs described above target a tumor antigen and the CD3 subunit of the TCR. Interestingly, several new BsAb formats are in development, including BsAbs redirecting specific T cell subsets, such as γδ T-cells, to the tumor [101,102]. LAVA-051 is a bispecific antibody that cross-links the Vδ2-TCR chain of Vγ9Vδ2-T cells (the most dominant population of γδ T cells [102] with the surface antigen CD1d present on MM cells [101]. This antibody effectively triggered the elimination of MM cells in vitro and in a mouse model [101]. This formed the rationale for the first-in-human dose-escalation study with LAVA-051 in heavily pre-treated MM and other CD1d-positive hematologic malignancies.

NK cell engagers are also being evaluated in cancer. These antibodies simultaneously target antigens on tumor cells and activating receptors on NK cells, resulting in strong interactions between both cell types and increased NK cell effector function [103]. NK cell engagers may be less likely to induce CRS, compared to T cell engagers. RO7297089 is an example of a bispecific NK cell engager that targets BCMA on MM cells and CD16a on NK cells. A dose-escalation study with RO7297089 is ongoing. Co-engagement of two NK cell receptors by trispecific NK cell engagers may further improve anti-tumor activity [103]. At this moment, there are no data available from clinical studies about the efficacy and tolerability of NK cell engagers. Importantly, NK cell numbers are substantially decreased during treatment with CD38 antibodies, as a result of high CD38 expression on NK cells [104], and it takes 3–6 months before NK cell frequency is restored to baseline levels [104]. This indicates that prior CD38 antibody therapy probably affects the efficacy of treatment with NK cell engagers.

## 5. Conclusions

In conclusion, BsAbs targeting MM-associated antigens have shown promising activity and efficacy in both preclinical and clinical studies. BsAbs offer a novel approach for the treatment of triple-class or penta-drug refractory patients. We expect that BsAbs can be effectively and safely combined with other agents to further improve the depth of response and response duration. Altogether, BsAbs form a promising treatment platform and will transform MM treatment.

## Figures and Tables

**Figure 1 jcm-10-04593-f001:**
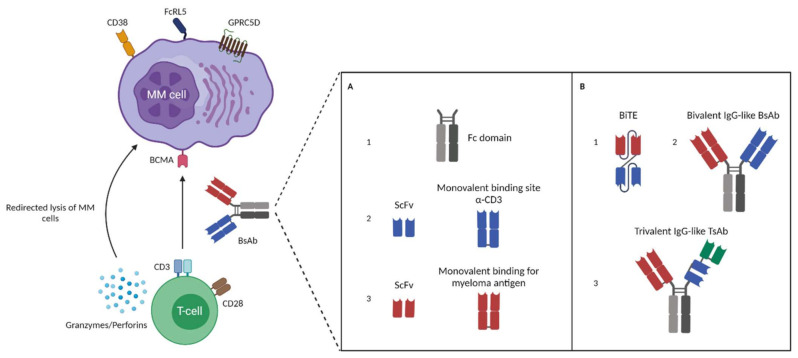
A schematic overview of different formats of bispecific antibodies used to initiate redirected lysis of multiple myeloma cells by T cells. Bispecific antibodies (BsAbs) bind simultaneously with one arm to CD3 expressed on T cells and with the other arm to a tumor-associated antigen (TAA) on the MM cell surface. This includes BCMA, CD38, FcRH5, and GPRC5D. The interaction leads to activation and degranulation of T cells (release of granzymes/perforins) and subsequent lysis of MM cells. (**A**,**B**) Next to the bivalent IgG-like BsAbs, bispecific T cell engagers (BiTEs) and trivalent IgG-like trispecific antibodies (TsAbs) are also able to mediate T cell-dependent lysis of MM cells. (**A**) (1) An Fc domain that connects two antigen-binding domains in IgG-like BsAbs and TsAbs. (**A**) (2) T cell binding domain that consists of an scFv with a monovalent binding site for CD3. (**A**) (3) Tumor cell-binding domain that consists of an scFv with a monovalent binding site for MM-associated antigen. (**B**) (1) A bispecific T cell engager (BiTE) that is comprised of two different scFvs connected with a peptide linker (e.g., AMG420). (**B**) (2) A bivalent IgG-like BsAb comprised of an Fc domain and two monovalent binding domains. (**B**) (3) A trivalent IgG-like TsAb comprised of an Fc domain and three monovalent binding domains including a binding domain for a third antigen (e.g., CD28).

**Figure 2 jcm-10-04593-f002:**
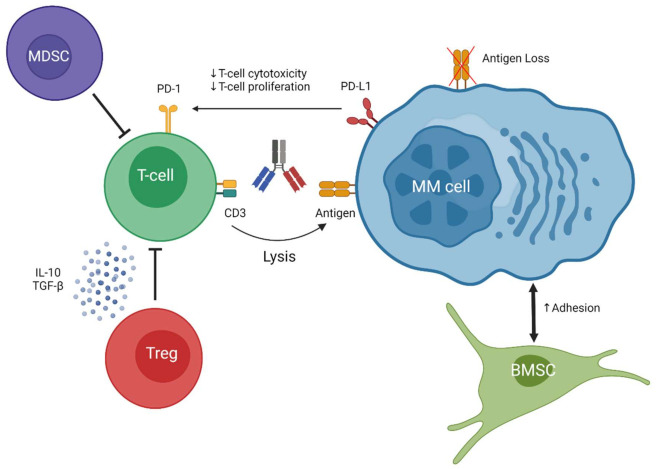
A schematic overview of different resistance mechanisms that impair BsAb-induced T cell-mediated lysis of MM cells. BMSC = Bone marrow stromal cell; MDSC = Myeloid-derived suppressor cell; MM cell = Multiple myeloma cell; Treg = regulatory T cell.

**Table 1 jcm-10-04593-t001:** An overview of clinical trials assessing T cell-redirecting bispecific antibodies in heavily pre-treated multiple myeloma patients.

Targets	Drug Name	Design	Trial Type (Phase)	Enrolled Patients	Overall Response Rate	Minimal Residual Disease	Cytokine Release Syndrome	Clinical Trial Identifier/Reference
BCMA x CD3	Pacanalotamab (AMG 420)	BiTE	1	42	31% (13 of 42 patients, IV); 70% (7 of 10 patients) at the MTD (400 µg/day)	71% (5 of 7 patients) at the MTD	38% (16 of 42 patients): Grade 1 = 28%; Grade 2 = 5%; Grade 3 = 2%	NCT02514239 [34]
BCMA x CD3	Pavurutamab (AMG 701)	Half-life extended BiTE (scFvs plus Fc domain)	1	85	26% (21 of 82 patients, IV); 83% (5 of 6 patients) at the most recent evaluable cohort	85% (6 of 7 patients) across all cohorts	65% (55 of 85 patients): Grade 1 = 27%; Grade 2 = 28%; Grade 3 = 9%	NCT03287908 [36]
BCMA x CD3	Elranatamab (PF-06863135)	IgG2a Fc region	1	30 (SC administration)	70% at SC doses ≥215 µg/kg (14 of 20 patients); 83% (5 of 6 patients) at the RP2D (1000 µg/kg)	100% (3 of 3 patients) across all cohorts	73.3%: Grade 1 = 57%; Grade 2 = 17% (no events >grade 2)	NCT03269136 [37]
BCMA x CD3	Teclistamab (JNJ-64007957)	IgG4 Fc region	1	157 total 40 at the RP2D	65% (26 of 40 patients) at the RP2D (SC 1500 µg/kg)	100% (6 of 6 evaluable patients treated at the RP2D); 69% (18 of 26 patients) across both IV and SC cohorts	70% (28 of 40 patients) at the RP2D: Grade 1 = 45%; Grade 2 = 25% (no events >grade 2)	NCT03145181 [38,39]
BCMA x CD3	REGN5458	VelociBi^TM^ Fc region	1	49	39% (19 of 49 patients, IV); 63% in dose-level 6 (8 patients)	57% (4 of 7 patients) across all cohorts	39% (19 of 49 patients): Grade 1 = 33%; Grade 2 = 6% (no events >grade 2)	NCT03761108 [40]
BCMA x CD3	REGN5459	*VelociBi*^TM^ Fc region	1	N/A	N/A	N/A	N/A	NCT04083534
BCMA x CD3	TNB-383B	IgG4 Fc region	1	58	46.5% (27 of 58 patients, IV); 80% (12 of 15 patients) at IV doses ≥ 40 mg	75% (3 of 4 patients) across all cohorts	80% (12 of 15 patients treated at IV doses ≥ 40 mg): Grade 1 = 46.7%; Grade 2 = 33.3% (no events >grade 2)	NCT03933735 [41]
BCMA x CD3	CC-93269	IgG1-based Fc region	1	30	43% (13 of 30 patients, IV); 89% (8 of 9 patients) at highest dose of 10 mg	92% (12 of 13 patients) across all cohorts	77% (23 of 30 patients): Grade 1 = 50%; Grade 2 = 23%; grade ≥3 = 3%	NCT03486067 [42]
CD38 x CD3	AMG424 *	Fc region	1	27	N/A	N/A	N/A	NCT03445663 [24]
CD38 x CD3	GBR 1342	BEAT^®^ platform	1	N/A	N/A	N/A	N/A	NCT03309111 [43]
CD19 x CD3	Blinatumomab **	BiTE	1	6	N/A	N/A	N/A	NCT03173430 [33]
FcRH5 x CD3	Cevostamab	IgG1-based Fc region	1	53	53% (18 of 34 patients, IV) at doses ≥3.6/20 mg	86% (6 of 7 evaluable patients) across all cohorts	76% (40 of 53 patients): Grade 1 = 18%; Grade 2 = 40%; Grade 3 = 2%	NCT03275103 [44]
GPRC5D x CD3	Talquetamab	IgG4 Fc region	Phase 1	Total of 184 patients with 30 participants treated at the RP2D	70% (21 of 30 patients) at the RP2D (SC 405 µg/kg, with 10.0 and 60.0 µg/kg step-up doses)	67% (4 of 6 patients) across both IV and SC cohorts including 1 patient from the RP2D cohort	73% (22 of 30 patients) at the RP2D: Grade 1 = 60%; Grade 2 = 10%; Grade 3 = 3%	NCT03399799 [45]

MTD = maximum tolerated dose; CRS = cytokine release syndrome; RP2D = recommended phase 2 dose; IV = intravenously administrated; SC = subcutaneously administered; N/A = not available; * Terminated due to sponsor business decision; ** Terminated due to slow accrual.

## Data Availability

Not applicable.

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
