# Peer review of "Current State of the Art and Prospects of T Cell-Redirecting Bispecific Antibodies in Multiple Myeloma"

_jcm, 2021, doi:10.3390/jcm10194593_

Round 1

Reviewer 1 Report

In this review Hosny et al. present the landscape of bispecific antibodies in multiple myeloma. The manuscript is very well-written and a pleasure to read.

Minor comments:

line 356: "on day eight"

line 356: "with as contributing factors MM progression" - please rephrase this sentence, it is confusing.

line 358: repeated paragraph of the discussion, please remove.

Congratulations to the authors.

Author Response

REVIEWER 1: Comments and Suggestions for Authors

In this review Hosny et al. present the landscape of bispecific antibodies in multiple myeloma. The manuscript is very well-written and a pleasure to read.

Minor comments:

line 356: "on day eight"

REPLY: We thank the reviewer for carefully reviewing our manuscript. We changed “on cycle one day eight” into “on day eight of the first cycle”.

line 356: "with as contributing factors MM progression" - please rephrase this sentence, it is confusing.

REPLY: We agree with the reviewer that this is unclear, as many patients are progressing when they are enrolled in the study. Therefore we changed “with as contributing factors MM progression with extensive extramedullary disease” into “with as contributing factors high tumor burden with extensive extramedullary disease”.

line 358: repeated paragraph of the discussion, please remove.

REPLY: we removed this sentence.

Reviewer 2 Report

The authors provide very comprehensive of the updated information on the immunotherapy in MM both mechansim of actions and detail in each study. Overall manscript is well-written, I have only minor comments.

1. It has a duplicated paragraph in the lines 532 - 543.

2. The authors mentioned in top of page 7 that the target antigen of BsAb should have important role on the tumor cell "In addition, the target antigen should be critical for the tumor cell so that down-regulation or complete loss of target expression will be disadvantageous for the tumor" However, the role of both FcRH5 and GPRC5D have not been determined, the effecicy of these 2 targeted BsAb demonstrated very promising efficacy. Does the efficacy of BsAb more depend on the effect of redirection of the T cell to lyse myeloma cell than blocking or down-regulation of the antigen on the tumor cell surface?

Author Response

REVIEWER 2: Comments and Suggestions for Authors

The authors provide very comprehensive of the updated information on the immunotherapy in MM both mechansim of actions and detail in each study. Overall manscript is well-written, I have only minor comments.

  1. It has a duplicated paragraph in the lines 532 - 543.

REPLY: We thank the reviewer for carefully reviewing our manuscript. As the reviewer pointed out, we removed the doubled paragraph.

  1. The authors mentioned in top of page 7 that the target antigen of BsAb should have important role on the tumor cell "In addition, the target antigen should be critical for the tumor cell so that down-regulation or complete loss of target expression will be disadvantageous for the tumor" However, the role of both FcRH5 and GPRC5D have not been determined, the effecicy of these 2 targeted BsAb demonstrated very promising efficacy. Does the efficacy of BsAb more depend on the effect of redirection of the T cell to lyse myeloma cell than blocking or down-regulation of the antigen on the tumor cell surface?

REPLY: We agree with the reviewer that the role of GPRC5D and FcRH5 in MM pathogenesis is still unknown. We also agree that a good target antigen should be stably expressed without downregulation or rapid development of antigen loss. Characteristics of the bispecific antibody are also important on the ability to effectively kill the MM cell and this includes the epitope that is targeted and the affinity of the binding interactions.

We therefore added on page 7 more details on characteristics of the target antigen and of the BsAb that may have an impact on the activity of the antibody. We added: “Target internalization may also have a negative impact on the activity of the BsAb, as this will limit immune synapse formation. In addition, activity of BsAbs is also affected by their molecular format, including binding affinity of each arm. Furthermore, for antigens with large extracellular domains, the epitope targeted by the antibody may influence the efficiency of immune synapse formation with BsAbs binding to the membrane proximal region having more efficient immune synapse formation, compared to BsAbs binding the central or distal domains”.